# Triple-Band Anisotropic Perfect Absorbers Based on α-Phase MoO_3_ Metamaterials in Visible Frequencies

**DOI:** 10.3390/nano11082061

**Published:** 2021-08-13

**Authors:** Bin Tang, Neigang Yang, Xianglian Song, Gui Jin, Jiangbin Su

**Affiliations:** 1School of Microelectronics and Control Engineering, Changzhou University, Changzhou 213163, China; csyng@icloud.com.cn (N.Y.); jbsu@cczu.edu.cn (J.S.); 2International Collaborative Laboratory of 2D Materials for Optoelectronic Science & Technology of Ministry of Education, Institute of Microscale Optoelectronics (IMO), Shenzhen University, Shenzhen 518060, China; songxianglian@outlook.com; 3Department of Electronic Information and Electronic Engineering, Xiangnan University, Chenzhou 423000, China

**Keywords:** perfect absorber, metamaterials, α-MoO_3_, polarization-dependent

## Abstract

Anisotropic materials provide a new platform for building diverse polarization-dependent optical devices. Two-dimensional α-phase molybdenum trioxides (α-MoO_3_), as newly emerging natural van der Waals materials, have attracted significant attention due to their unique anisotropy. In this work, we theoretically propose an anisotropic perfect metamaterial absorber in visible frequencies, the unit cell of which consists of a multi-layered α-MoO_3_ nanoribbon/dielectric structure stacked on a silver substrate. Additionally, the number of perfect absorption bands is closely related to the α-MoO_3_ nanoribbon/dielectric layers. When the proposed absorber is composed of three α-MoO_3_ nanoribbon/dielectric layers, electromagnetic simulations show that triple-band perfect absorption can be achieved for polarization along [100], and [001] in the direction of, α-MoO_3_, respectively. Moreover, the calculation results obtained by the finite-difference time-domain (FDTD) method are consistent with the effective impedance of the designed absorber. The physical mechanism of multi-band perfect absorption can be attributed to resonant grating modes and the interference effect of Fabry–Pérot cavity modes. In addition, the absorption spectra of the proposed structure, as a function of wavelength and the related geometrical parameters, have been calculated and analyzed in detail. Our proposed absorber may have potential applications in spectral imaging, photo-detectors, sensors, etc.

## 1. Introduction

Two-dimensional (2D) materials with weak van der Waals (vdW) interaction between atomic layers have drawn wide attention due to their extraordinary physical, chemical, and optoelectronic properties, such as graphene [1], hexagonal boron nitride (h-BN) [2,3], transition metal dichalcogenides (TMDs) [4], and black phosphorus (BP) [5,6,7,8]. By breaking the vdW bonds, monolayer 2D materials can be achieved and further transferred to the desired substrates [9,10,11]. In recent years, it has been demonstrated, both theoretically and experimentally, that plasmons can support low propagation loss and strong field confinement in a few or even monolayer 2D materials. Additionally, the spectral responses in these 2D materials can be dynamically tuned by chemical doping or electrical bias. Recently, a new type of 2D vdW semiconducting crystal, α-phase molybdenum trioxide (α-MoO_3_), has intrigued many researchers due to its highly anisotropic properties [12]. The strong anisotropy, originating from the unique crystalline structure of α-MoO_3_, can lead to previously reported naturally occurring in-plane hyperbolicity that is associated with the distinct lattice modes along three orthogonal crystal directions. In contrast to the monoclinic and hexagonal phase of crystal MoO_3_, orthorhombic MoO_3_ (α-phase) is more thermodynamically stable, which is constituted by planar layered sheets of linked and distorted MoO_6_ octahedra. Therefore, the newly emerging α-MoO_3_ materials can be integrated into metamaterials to achieve more freedom in controlling light–matter interactions. Moreover, it provides a new opportunity in the applications of polarization angle-dependent photonic devices, including polarization reflectors and polarization color filters [13]. In addition, some other novel physical phenomena and applications were found based on α-phase molybdenum trioxide. For instance, Hu et al. observed the topological polaritons and photonic magic angle in twisted α-MoO_3_ bi-layers [14]. Dereshgi et al. proposed lithography-free IR polarization converters via orthogonal in-plane phonons in α-MoO_3_ flakes [15].

As an important application in metamaterials, a perfect absorber [16,17,18,19,20], which is constructed by metal to achieve resonant coupling with incident electromagnetics, was first proposed by Landy et al. [21]. Since then, narrow-band absorption [22,23], broadband absorption [24,25], chiral-selective absorption [26], polarization-independent and angle-insensitive tunable absorption [27], and even multi-band metamaterial absorbers [28,29] have been extensively studied and applied in plasmonic light harvesting [30], polarization detectors [31], solar energy [27], and infrared cloaking [32]. In recent years, a variety of 2D-materials-based metamaterials have been proposed for achieving tunable enhancing optical absorbance and perfect absorption. For example, Luo et al. theoretically proposed a tunable narrowband absorber based on graphene film and realized an ultra-narrowband perfect absorption peak [33]. Zhu et al. theoretically proposed a tunable ultra-broadband and wide-angle perfect absorber based on stacked monolayer black phosphorus nanoribbons and black phosphorus–dielectric–metallic hybrid architecture [34,35], respectively. Sang et al. presented a dual-band absorber based on monolayer molybdenum disulfide (MoS_2_) [36]. Recently, Deng et al. presented a broadband absorber with an in-plane trapezoid α-MoO_3_ patch arrays structure in an infrared regime [37]. Dong et al. proposed an absorber with a single resonant peak by utilizing α-MoO_3_ combined with a dielectric distributed Bragg reflector in a mid-infrared frequency [38]. However, the potential of α-MoO_3_ materials in metamaterial absorbers has been underutilized. In particular, to our knowledge, an α-MoO_3_-based perfect absorber with multiband in the visible frequency regime has not yet been reported. Our effort in this work is directed toward extending single-band perfect absorption to triple-band absorption, which is of great importance for wide applications, such as spectral imaging, photodetectors, sensors, etc.

In this work, we theoretically and numerically propose an anisotropic metamaterial absorber which is composed of multiple α-MoO_3_/dielectric layers stacked on a silver mirror. Triple-band perfect absorption can be achieved in visible frequencies for both polarizations along *x*- and *y*-directions, respectively. The highly anisotropy results from the anisotropic lattice structure and the polarization-dependent complex refractive index of α-MoO_3_. Additionally, the operation mechanism of perfect absorption can be explained by the resonant grating modes and the interference effect of Fabry–Pérot cavity modes. Moreover, simulation results obtained by the finite-difference time-domain (FDTD) method are consistent with the theoretical analysis of impedance match. In addition, the absorption spectra of the proposed structure, as a function of wavelength and the related geometrical parameters, have been calculated and analyzed in detail.

## 2. Materials and Methods

Figure 1a schematically shows the unit cell of the proposed perfect absorber consisting of alternating layers of α-MoO_3_ nanoribbon and dielectrics stacked on a silver mirror. The related geometrical parameters and their corresponding values are given in the caption of Figure 1. In the design, the crystal directions [100], [001], and [010] refer to *x*-, *y*-, and *z*-directions in the rectangular coordinate system, separately. Figure 1b illustrates the orthorhombic α-MoO_3_ structure, which has three different types of oxygen atom, i.e., terminal O_t_ along [010] direction bonded with a Mo atom, asymmetric Oa along [100] direction bonded with two Mo atoms, and symmetrical Os along [001] direction bonded with three Mo atoms. In general, the special atomic structure highly influences the properties of α-MoO_3_, resulting in unique optical anisotropy. Here, we use complex dielectric function to describe its optical properties [13]:(1)$${\varepsilon (\omega )=\varepsilon }_{\infty }+\underset{j}{\sum }\frac{\omega_{pj}^{2}}{\omega_{oj}^{2}{-\omega }^{2}{-i\gamma }_{j}\omega }$$
where $i=\sqrt{-1}$, *j* indicates the total number of oscillators, *ε*
_∞_, *ω_pj_*, *ω_oj_*, and *γ_j_* refer to the high frequency dielectric constant, the plasma frequency, the eigenfrequency, and the scattering rate of the *j*th Lorentz oscillator, respectively. The above parameters used in Equation (1) are given in Table 1. Figure 1c,d shows the real and imaginary part of the complex refractive index of α-MoO_3_ in the visible region. Compared with the other 2D materials, such as graphene and BP, the complex refractive index of α-MoO_3_ exhibits a good trade-off between high index and low loss.

In simulations, the thickness of α-MoO_3_ was assumed to be 10 nm. The full-field electromagnetic wave calculations were performed by a three-dimensional FDTD method. Additionally, the light waves were incident from the top side along the negative *z*-direction, and periodic boundary conditions were applied in *x*- and *y*- directions. The spectral absorption is calculated by *A* = 1 − *R* − *T*, where *R* and *T* represent the reflection and transmission, respectively. Considering the existence of thick silver layer (200 nm), the transmission *T* is zero. Therefore, the absorption can be simplified by *A* = 1 − *R*. In addition, impedance of the proposed structure should match with that of normalized impedance in free space (*Z* = *Z*
_0_ = 1) under critical coupling conditions. The effective impedance of the system can be expressed as [39]:(2)$$Z=\sqrt{\frac{{\left({1+S}_{11}\right)}^{2}{-S}_{21}^{2}}{{\left({1-S}_{11}\right)}^{2}{-S}_{21}^{2}}}$$
(3)$$S_{11}=\frac{i}{2}\left(\frac{1}{Z}-Z\right)sin\left(nkd\right)$$
(4)$$S_{21}=\frac{1}{cos\left(nkd\right)-\frac{i}{2}\left(Z+\frac{1}{2}\right)sin\left(nkd\right)}$$
where *S*
_11_ is the scattering parameters and *S*
_21_ is the transmission coefficients, respectively. *n*, *k*, and *d* are the effective refractive index, the wave vector, and the thickness of the designed structure, separately. Furthermore, the dielectric layer is assumed to be SiO_2_ with a refractive index of 1.45, and the permittivity of silver is obtained from Drude mode [35]: *ε_m_*(*ω*) = *ε_∞_* − *ω_p_*
^2^/(*ω*
^2^ + *iωγ*), where *ω* is the angular frequency, the plasma frequency is *ε_p_* = 1.39 × 10^16^ rad/s, the scattering rate is *γ* = 2.7 × 10^13^ rad/s, and the high-frequency constant is *ε*
_∞_ = 3.4.

## 3. Results and Discussion

Figure 2a shows the simulated optical absorption spectra for the structure composed of a single layer α-MoO_3_ nanoribbon with a width *w*_1_ = 105 nm on a silver substrate when illuminated by polarized lights along [100] and [001] directions, respectively. It can be found from Figure 2a that two resonant peaks exist for both polarizations. When the incident light is polarized along the [100] direction, perfect absorption is achieved at *λ*
_11_ = 657.4 nm. One can conclude from the *z*-components of electrical field in Figure 2a that the perfect absorption peak mainly originates from the resonant excitation of grating modes, which is caused by periodicity, as explained in reference [40]. By contrast, weak resonant absorption at the wavelength of *λ*
_12_ = 569.2 nm is due to the excitation of the localized surface plasmons on the interface of metal Ag because of the existence of α-MoO_3_ nanoribbon grating. In addition, the resonant absorption peak has a blue-shift when polarization is switched to the [001] direction of α-MoO_3_. Interestingly, when a second α-MoO_3_ nanoribbon with a width *w*
_2_ = 275 nm is stacked on the first α-MoO_3_ nanoribbon separated by a dielectric layer, as shown by the inset in Figure 2b, dual-band perfect absorption can be established for both polarization directions due to the absorption enhancement of the weak resonant absorption peak at the short wavelengths.

Based on the above analysis, one can expect that more perfect absorbing bands could be obtained with multiple stacking of α-MoO_3_ nanoribbon/dielectric layer. As displayed in Figure 3a, triple-band perfect absorption with recognizable anisotropy is observed by three stacking layers of α-MoO_3_ nanoribbon/dielectric on Ag substrate. Specifically, when the incident light is polarized along [100] direction of α-MoO_3_, three absorbing peaks locate at *λ*
_1_ = 660.1 nm, *λ*
_2_ = 601.5 nm, and *λ*
_3_ = 515.2 nm, with corresponding absorptivity up to 99.5%, 99.7%, and 99.6%, separately. Moreover, the effective impedance of such an absorber is calculated according to Equation (2), as shown in Figure 3b. When the polarization is along [100] direction of α-MoO_3_, the corresponding effective impedances at the three resonant absorption peaks are *Z*
_1_ = 0.996 − 0.152i, *Z*
_2_ = 0.988 − 0.124i, and *Z*
_3_ = 1.081 + 0.021i, separately. Obviously, all the effective impedances match quite well with the normalized impedance of free space, which indicates that optical reflection is suppressed effectively, thus resulting in the triple-band optical perfect absorption. In addition, three resonant absorption peaks also exist, and they remain in perfect absorbance while switching the polarization incidence along the [001] direction of α-MoO_3_. Meanwhile, all the resonant peaks shift to the shorter wavelengths due to the anisotropic lattice structure. The corresponding effective impedances at the three resonant absorption peaks are *Z*
_1_ = 0.985 − 0.084i, *Z*
_2_ = 1.127 − 0.124i, and *Z*
_3_ = 1.251 + 0.013i, separately. Additionally, the real and imaginary parts of the effective impedances of those three peaks under such a circumstance match well with those of the free space, as shown in Figure 3c.

To interpret the underlying physics mechanism behind the multi-band perfect absorbing phenomenon, we calculate, in Figure 3d–f, the field distributions on the *x*–*z* plane at the different resonant frequencies, i.e., *λ*
_1_ = 660.1 nm, *λ*
_2_ = 601.5 nm, and *λ*
_3_ = 515.2 nm, for polarization along the [100] direction of α-MoO_3_. From Figure 3d–f, one can see that the localized electric field is enhanced at the edges of α-MoO_3_ nanoribbons for each layer, which corresponds to the different resonant responses. In particular, at the wavelength of *λ*
_1_ = 660.1 nm, the electric field hot spot is mainly excited in the bottom α-MoO_3_ nanoribbon, and the resonant wavelength is roughly consistent with *λ*
_11_ = 657.4 nm in a one-layer α-MoO_3_ nanoribbon structure, which is referred to as grating mode. However, at the wavelengths of *λ*
_2_ = 601.5 nm and *λ*
_3_ = 515.2 nm, electric energies are strongly concentrated on the edges of middle and top α-MoO_3_ nanoribbons, respectively. For the both cases, the stacking α-MoO_3_/dielectric layers on a silver mirror form the Fabry–Pérot cavity, and the resonant absorption mainly results from the cavity modes’ interference effect. The incident light is reflected back and forth between the α-MoO_3_ nanoribbons and metal substrate, with a complex propagation phase $\tilde{\beta }=\sqrt{\tilde{\varepsilon }}k_{0}h$, where $\tilde{\varepsilon }$ is the permittivity of the spacer, *k*
_0_ is the wavenumber in free space, and *h* is the dielectric thickness. The absorbance can be derived from $A\left(\omega \right)=1-{\left|\tilde{r}\left(\omega \right)\right|}^{2}$, where $\tilde{r}\left(\omega \right)$ is the total reflection resulting from the superposition of multiple reflections. Obviously, spacer thickness *h* plays a major role. Figure 4a,b depicts the absorption spectra of the proposed structure as a function of wavelength and the thickness of SiO_2_ dielectric layer. It is clearly seen from Figure 4 that the dielectric thickness directly affects the resonance wavelength of the Fabry–Pérot cavity. With the increase in SiO_2_ layer thickness, all three resonance peaks shift to a long wavelength. The maximum absorbance occurs at the constructive interference, with a phase condition of $2\tilde{\beta }+\phi +\pi \approx 2m\pi$, where *ϕ* represents the reflection phase shift and *m* is an integer.

Figure 5 illustrates the optical absorption as a function of incident wavelength and widths of the three α-MoO_3_ nanoribbons. The width ratio constant with *w*_3_*/w*_2_ = 1.3, *w*_2_*/w*_1_ = 2.6, respectively. While increasing the width of the nanoribbons, the distance of the nanoribbons in the adjacent period shortens and the interaction increases; all the resonant absorbing peaks exhibit a minor shifting to the longer wavelengths, and absorptivity is higher than 90%. The reason can be attributed to the fact that the wider α-MoO_3_ nanoribbons support the electromagnetic responses at longer wavelengths. Besides, Figure 6 shows the influence of periodicity on the absorption spectra for both [100] and [001] polarizations. Apparently, all the resonant peaks shift toward the longer wavelength while increasing the periodicity of the α-MoO_3_ absorber. As the period becomes larger, the distance between adjacent α-MoO_3_ nanoribbons enlarges as well. As a result, the interaction between them creates a change, thus leading to a shifting of resonant frequencies. In short, it can be concluded that a multi-band α-MoO_3_-based perfect absorber in visible frequencies can be optimized by changing the geometrical parameters of the structure.

## 4. Conclusions

In conclusion, we theoretically propose and numerically demonstrate an anisotropic perfect metamaterial absorber in visible frequencies; it consists of a multi-layered α-MoO_3_/dielectric structure stacked on a silver mirror. Additionally, the number of perfect absorption bands is closely related to the α-MoO_3_ nanoribbon/dielectric layers. When the proposed absorber is composed of three α-MoO_3_ nanoribbon/dielectric layers, the electromagnetic simulations show that triple-band perfect absorption can be achieved for polarization along the [100] and [001] directions of α-MoO_3_, respectively. Additionally, the calculation results obtained by FDTD method are consistent with the effective impedance of the designed absorber. The physical mechanism of the perfect absorption can be attributed to the resonant grating modes and the interference effect of Fabry–Pérot cavity modes. In addition, we have calculated and analyzed the absorption spectra of the proposed structure as a function of wavelength and the related geometrical parameters in detail. Moreover, our method can be flexibly extended to obtain a tunable multi-band perfect absorber in the visible frequencies by adjusting the numbers of layers in the α-MoO_3_/dielectric stacking structure. Our research opens up an avenue for designing anisotropic meta-devices with tunable spectra, and it has potential applications in spectral imaging, photodetectors, and sensors.

## Figures and Tables

**Figure 1 nanomaterials-11-02061-f001:**
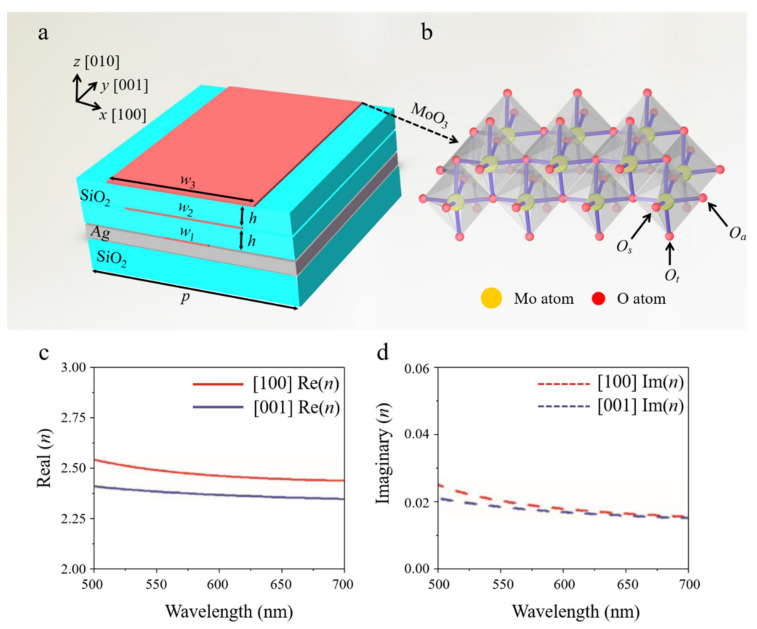
(**a**) The schematic diagram of the unit of proposed absorber, consisting of multilayer α-MoO_3_ nanoribbons/SiO_2_ stacked on a silver substrate. In this design, the optimized geometrical parameters are listed as follows: *h* = 100 nm, *w*
_1_ = 105 nm, *w*
_2_ = 275 nm, *w*
_3_ = 345 nm, *p* = 500 nm. (**b**) The layered orthorhombic α-MoO_3_ structure, in which the yellow and red balls represent molybdenum and oxygen atoms, respectively. Real (**c**) and imaginary (**d**) parts of the refractive index of α-MoO_3_ in the visible region.

**Figure 2 nanomaterials-11-02061-f002:**
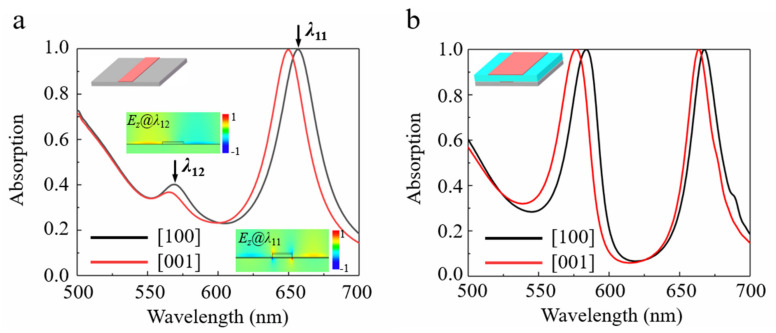
(**a**) Simulated absorption spectra of the α-MoO_3_ nanoribbon stacked on Ag substrate, as polarized light along [100] (black curve) and [001] directions (red curve), respectively. (**b**) Simulated absorption spectra for the structure composed of Ag substrate/α-MoO_3_ nanoribbons/SiO_2_ dielectric layer/α-MoO_3_ nanoribbons from bottom to top, as polarized along [100] (black curve) and [001] (red curve) directions, respectively. The optimized geometrical parameters are listed as follows: *w*
_1_ = 105 nm, *w*
_2_ = 275 nm, *h* = 100 nm, *p* = 500 nm.

**Figure 3 nanomaterials-11-02061-f003:**
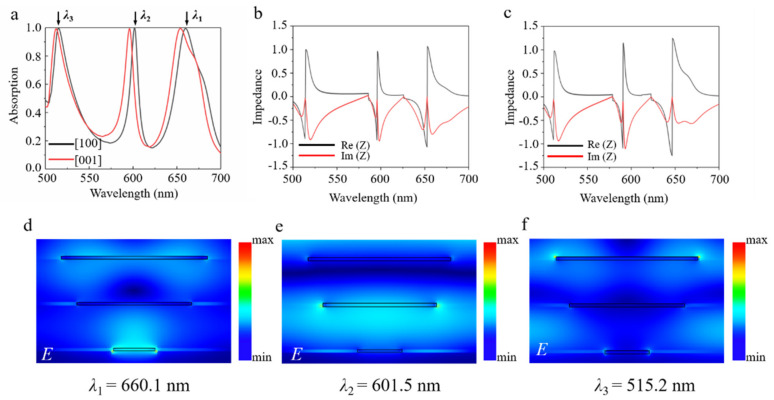
(**a**) Simulated absorption spectra for the proposed triple layer α-MoO_3_ structure along [100] and [001] directions, respectively. (**b**,**c**) The calculated real and imaginary parts of the effective impedance along the [100] and [001] directions, respectively. The optimized geometrical parameters are listed as follows: *h* = 100 nm, *w*
_1_ = 105 nm, *w*
_2_ = 275 nm, *w*
_3_ = 345 nm, *p* = 500 nm. (**d**–**f**) The calculated distribution of total electric field at the resonant wavelengths of *λ*
_1_ = 660.1 nm, *λ*
_2_ = 601.5 nm, and *λ*
_3_ = 515.2 nm, separately.

**Figure 4 nanomaterials-11-02061-f004:**
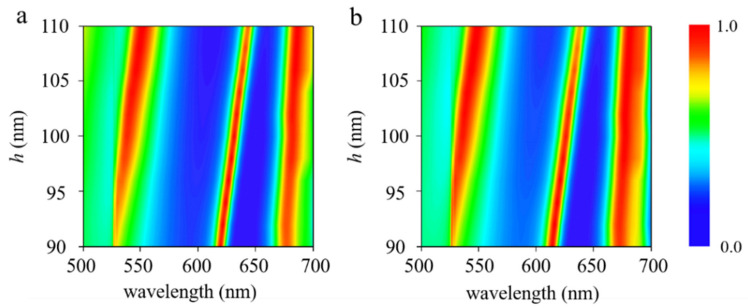
Absorption spectra of the proposed structure as a function of wavelength and the height of SiO_2_ dielectric layer *h*, with polarization along (**a**) [100] and (**b**) [001] crystalline directions of α-MoO_3_.

**Figure 5 nanomaterials-11-02061-f005:**
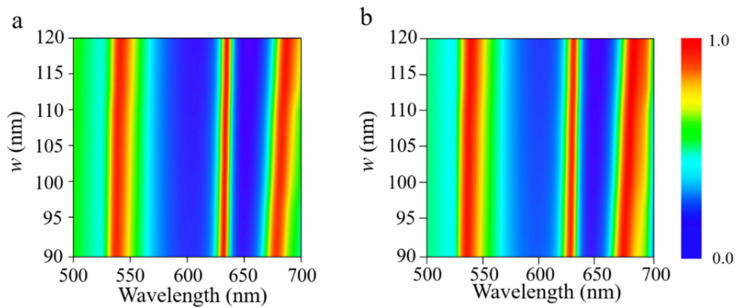
Absorption spectra of the proposed structure as a function of wavelength and the width *w* of α-MoO_3_ nanoribbons with polarization along (**a**) [100] and (**b**) [001] crystalline directions of α-MoO_3_; the width ratio constant is kept unchanged.

**Figure 6 nanomaterials-11-02061-f006:**
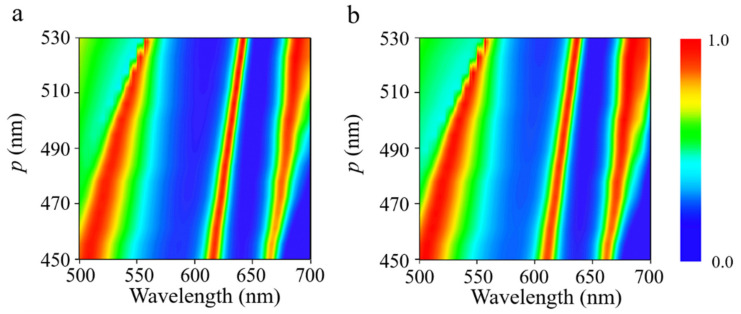
Absorption contour of the proposed structure as a function of wavelength and periodicity *p* of the proposed structure for incident polarization along (**a**) [100] and (**b**) [001] crystalline directions of α-MoO_3_.

**Table 1 nanomaterials-11-02061-t001:** Parameters used in Equation (1) to obtain the permittivity tensors of α-MoO_3_ in the visible range.

Polarization	*ε* _∞_	*ω_pj_* [cm^−1^]	*ω*_0_ [cm^−1^]	*γ_j_* [cm^−1^]
*x*	5.065	21,672	27,019	1342.2
*y*	4.502	29,078	32,271	2027.1

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
