# Peer review of "Triple-Band Anisotropic Perfect Absorbers Based on ?-Phase MoO3 Metamaterials in Visible Frequencies"

_nanomaterials, 2021, doi:10.3390/nano11082061_

Round 1

Reviewer 1 Report

They propose a triple-band anisotropic metasurface perfect absorber, which consists of three-layer α-MoO3/dielectric structure stacked on a silver mirror.

The paper is suitable for publication after the following issue

1) in the formula 1 the index i should be confused with the imaginary part i.

2) They present the formula (2) for the retrieval of the impedance, but they do not show any numerical results.

3) They should be discussed the chosen of the thickness of the layer, and how are linked to the maximum absorption

Reviewer 2 Report

I have the following suggestions to improve the paper quality so that it can be publishable in the Nanomaterials journal.  

  • Line 23, “such as” is written twice.
  • The author said, “Our effort in this paper is directed toward extending single-band perfect absorption to multi-band absorption spectrum, which is of great importance for wide applications of α-MoO3.” Please give examples of which applications?
  • What is the substrate used in the design? Please draw the substrate in figure 1.
  • Line 98, what are S21 and S11? Are they scattering parameters or transmission and reflection coefficient, respectively? Please clarify.
  • Justify the selection of silver layer? Isn’t it oxidizing in the air and non-biocompatible?
  • In Figures 4,5 and 6, the author has studied the effect of geometric parameters variation on the resonance wavelength. I suggest the author calculate the effect on absorption as well as the variation in geometric parameters.
  • What is the effect of angle of incidence on the absorption and resonance wavelength? Please give a graph.
  • Please explain the potential applications of the proposed device. The operational wavelength range is visible, how it can be used in sensing applications?
  • The introduction section is weak. The author has completely missed the previous literature related to multiband MS PAs such as https://doi.org/10.1364/OE.26.007066, https://doi.org/10.1142/S0217984917503547. And some recent attractive designs of narrowband MS PAs such as https://doi.org/10.1088/2040-8986/abf890. And suggest that why the multiband approach which you have proposed is better than the previous multiband MS PAs.
  • In the paper, the author has designed 3 absorption bands. Out of curiosity, I would like to know how many bands can be obtained by staking MoO3 nanoribbon gratings? Is there a limit?
  • Why has the author selected the visible spectrum for the device design? Can it be designed for other wavelength ranges?
  • Abstract and Conclusion sections need to be modified and provide the main results obtained in the paper such as absorption and the resonance wavelength, etc.

Round 2

Reviewer 2 Report

Accept in current form.